# Effect of Dietary Soy Protein Source on Effluent Water Quality and Growth Performance of Rainbow Trout Reared in a Serial Reuse Water System

**DOI:** 10.3390/ani13193090

**Published:** 2023-10-03

**Authors:** Thomas L. Welker, Ken Overturf

**Affiliations:** USDA-ARS, Hagerman Fish Culture Experiment Station, 3059-F National Fish Hatchery Road, Hagerman, ID 83332, USA; ken.overturf@usda.gov

**Keywords:** soybean meal, soy protein concentrate, raceway, effluent, rainbow trout, gene expression

## Abstract

**Simple Summary:**

Fishmeal is a finite resource that continues to serve as the primary protein source in feeds of salmonids, including rainbow trout. Partial or total replacement of fishmeal has been successful, but many of these substitutions are of plant origin containing anti-nutritional factors that negatively impact growth and limit their dietary inclusion level. Soybean meal and processed soy have been the primary alternative protein sources utilized in rainbow trout feeds. Even at acceptable levels for normal growth, soy can cause a diarrhea-like condition that leads to poor water quality, but these impacts have not been evaluated in a commercial production setting. We showed that the use of soy protein concentrate promotes growth similar to a fishmeal diet and is superior to soybean meal under these conditions. However, soy protein concentrate led to lower water quality for some parameters and suggests that there are other considerations aside from growth when evaluating the practicality of fishmeal replacement.

**Abstract:**

Juvenile rainbow trout (125 ± 0.8 g) were fed a fishmeal control diet (C), a high soy protein concentrate diet (SP), a high soybean meal diet (HiS), or a diet with high levels of fermented soy protein concentrate (fSP) for 12 weeks in a tank system capable of receiving 1st and 3rd use water from a serial-reuse production hatchery. Water quality was generally lower in 3rd use compared to 1st use water and after passing through tanks (inflow vs. outflow). Total dissolved solids were significantly higher (*p* = 0.003) for 3rd use compared to 1st use water, and values were also higher (*p* < 0.001) for the fSP diet. Turbidity and ammonia were highest in tanks for trout fed the HiS and fSP and SP and fSP diets, respectively, but were characterized by high variation, which likely prevented the detection of significant differences. Weight gain (*p* < 0.001) and survival (*p* = 0.008) were significantly lower for trout in 3rd use compared to 1st use water. Trout fed the HiS diet were generally in poorer physiological condition with lower body fat stores (*p* = 0.05) and lower growth rate (*p* < 0.001) and survival (*p* = 0.05) compared to the other diets, which were similar. The expression of several stress-associated genes (FK506, DIO2, REGPS, Cyp1a, G6PH, GADD45a, and IRF-1) in the liver and gill showed that diet and water source affected their regulation. Replacement of FM by SP providing 50% of dietary protein promotes acceptable growth performance compared to an FM diet and was superior to HiS. The impacts of soy protein concentrate on water quality under commercial production conditions, however, require further study.

## 1. Introduction

Fishmeal (FM) is the primary protein source utilized in commercial diets of rainbow trout. It is a complete protein source and highly digestible to trout [1]. However, it is a finite resource, and global demand for seafood is predicted to grow substantially in the coming years [2], leading to increased demand and associated higher costs for FM [3]. In response, trout growers and feed producers have searched for more economical and stable alternatives as full or partial replacements for FM. Among these replacements, soybean meal (SBM) has emerged as the most commonly adopted alternative protein source and successfully incorporated into the diets of rainbow trout and other fish species [1,4]. However, there are limitations associated with the use of SBM and other plant-based protein sources within fish feed formulations.

Soybean meal has high levels of structural fiber and anti-nutritional factors (ANF), such as protease inhibitors, lectins, phytic acid, saponins, phytoestrogens, and anti-vitamins, that limit its incorporation into diets of rainbow trout [5,6,7]. Soybean meal can also cause hindgut inflammation (enteritis), reduced appetite, and protease inhibition [5]. Inclusion rates of SBM in rainbow trout diets are typically kept at less than 20% [8]. Levels greater than 20% can cause reduced weight gain and feed efficiency and increased incidence of intestinal enteritis [9,10,11,12,13,14]. The negative impacts of ANF on trout health and performance must be taken into account when formulating diets with elevated levels of SBM; however, physical and chemical alterations can be used to reduce the effects of these anti-nutrients and increase the levels of soy protein sources in fish feeds.

Modification of soy protein products by chemical, mechanical, and biological methods can improve the nutritive value to fish [15]. These processes can be used to remove fiber and other anti-nutrients and fractions of low nutritive value to produce high-protein soy concentrates (SPC) [16,17,18,19]. Fermentation of soy products can also result in improved nutrient profiles by removing non-nutritive (e.g., oligosaccharides) or anti-nutritive (e.g., trypsin inhibitor and phytic acid) components [15,20,21,22,23,24]. The use of SPC has allowed a higher proportion of the crude protein in rainbow trout diets to be supplied by soy [15]). However, the addition of SPC to the diets of rainbow trout can cause increased excretion of ammonia and urea [11] and reduction of fecal stability with an associated increase in suspended fine fecal particles [25,26]. Little examination of the effects of SPC on water quality and their interactive effects on rainbow trout (RBT) performance has occurred, and more is required.

The environmental impact of aquaculture on receiving waters has come under increased scrutiny in recent years [27,28,29,30]. Most states require routine monitoring of phosphorus discharge from aquaculture facilities, and as the allowable discharge levels are decreasing, these constraints are limiting aquaculture production and the potential growth of the industry. Federal [31] and State [29] agencies are also beginning to monitor levels of nitrogen (ammonia) and suspended solids in addition to phosphorus released from aquaculture facilities. Elevated levels of dietary SBM can cause alterations in the distal intestinal epithelium, leading to pathological changes characterized by diarrhea. This can result in the production of fecal matter composed of extremely fine particles that do not settle within the quiescent zone of raceways, lowering water quality [32,33]. Reduced water quality has the potential to amplify stress loads and negatively impact the physiological performance of rainbow trout, leading to reduced growth during prolonged exposure to chronic stressors [34]. Conditions such as low dissolved oxygen, elevated ammonia levels, and suspended solids can serve as persistent stressors for rainbow trout [30,35]. In serial-reuse and serial passage production, water quality often deteriorates as the number of raceway passages multiplies and is characterized by an increase in ammonia and dissolved solids and lower dissolved oxygen [30,36,37]. Different dietary components, such as changes from fishmeal to soybeans as a protein source, are also known to cause a shift in the expression level of genes linked to stress [38]. The consequences of water quality stress have been examined through comprehensive analyses of global gene expression by RNA sequencing and through the evaluation of expression of known stressor genes [36,39,40,41]. We chose genes that were identified as affected by water quality and other fish culture stressors across various tissues, including gill and liver. Some participate in immunomodulation (FK506 and IRF-1), others in metabolic responses and cellular homeostasis including oxidative stress (Cyp1a, G6PH, and DIO2), and some that are multifunctional and sensitive to environmental stress (REGPS and GADD45a).

As the use of plant protein sources for total or partial replacement of FM becomes more common, concerns regarding the impact of these products on downstream water quality of the effluent of aquaculture production systems will need to be addressed [42]. To adequately determine the effects of feeds that contain elevated levels of plant products on downstream receiving waters, systems that can evaluate incoming water quality parameters and effluent are required. No research has examined the effect of soy protein sources on water quality in systems designed to be similar to production conditions, and there has been little examination of the interaction of water quality and dietary soy protein on trout performance. Research under commercial conditions is needed to determine what the impact of these products are and what changes are needed regarding trout production and feed formulation to reduce their effects on water quality. This information will be directly applicable to commercial trout production and is essential for continued and increased incorporation of soy into rainbow trout diets. The goal of this study was to examine the effects and interaction of dietary protein sources (three soy protein sources, SBM, SP, fSP; and FM) and water source (with varying water quality) on growth, stress response, and water quality.

## 2. Materials and Methods

### 2.1. Experimental Tank System and Fish Stocking

The Snake River Farm is a commercial trout production facility situated in the south-central region of Idaho, along the middle portion of the Snake River. Water is supplied to the farm by the Eastern Snake Plain Aquifer and operates as a single-pass, flow-through raceway system. The typical flow rate is approximately 0.17 m^3^/s, and water passes through five raceways before being discharged into the quiescent zone [43]. The initial water quality in these raceway systems is defined by the source spring water (1st use), with subsequent passes through raceways reducing water quality.

The Trout Grains Research group (USDA, Agriculture Research Service) in Hagerman, Idaho, has a research tank system at Snake River Farm that utilizes water directly from flow-through raceways. This system is designed so that banks of tanks are available to different water sources from raceways, which represent water conditions at various stages of commercial production; the system is also designed so that the incoming water and the effluent from each tank can be continuously analyzed for pH, turbidity, ammonia, and other water quality parameters. Using this system, we examined the dietary inclusion of three soy protein sources (soybean meal, soy protein concentrate, and a fermented soy protein concentrate) and FM (control) on water quality, stress response, and growth.

This project was conducted using 24 outdoor tanks (400 L) designed to receive water from the production raceways at Snake River Farm, Clear Springs Foods, Inc. (now Riverence, Inc.), located in Buhl, ID. Twelve tanks were supplied with 1st use water directly sourced from the spring, and another 12 tanks received 3rd use water, having passed through two production scale raceways of trout. Juvenile rainbow trout were collected from a raceway at Snake River Farm, subjected to tricaine methanesulfonate (MS-222) anesthesia (at a concentration of 50 mg/L), and then moved to the tank system by fish transport tank with supplementary oxygen. The trout were subsequently sorted and weighed, and batches of 30 fish, with an average weight of 125 ± 0.8 g, were randomly stocked into each tank. Tank conditions were regulated to approximate the conditions found in the Snake River Farm raceways. This involved maintaining an approximate stocking density of 25 kg/m^3^ and ensuring a water turnover rate of 3 times per hour (equivalent to 22.7 L/min) throughout the study period. Water entering each bank of 12 tanks (supply line; one for each water source) and exiting each individual tank was monitored using water quality sonde/data loggers equipped with sensors for temperature, dissolved oxygen, ammonia/ammonium levels, pH, and turbidity (Model 6920 V2; YSI, Inc., Yellow Springs, OH, USA). Water quality measurements were monitored every 3 weeks by recording for 24 h in each tank until all tanks were measured. Total dissolved solids (TDS) were calculated from conductivity readings using gravimetric measurement of dissolved matter after filtration and evaporation to produce a conversion factor according to EPA Standard Methods #160.1 [44]. Temporal water quality parameters within treatments did not vary, so values were averaged and reported for the entire study for each dietary treatment-water source combination.

### 2.2. Experimental Feeds and Feeding

Four soy protein sources, soybean meal (HiS; ADM, Decatur, IL), Profine VF soy protein concentrate (SP; The Solae Company, Fort Wayne, IN, USA; now Bunge, Inc.), and HP-300 fermented soy protein concentrate (fSP; Hamlet Protein, Inc., Findlay, OH, USA), were used to formulate experimental diets, each supplying 50% of the protein in diet (Table 1). A fishmeal control diet (C), supplying 50% of dietary protein (Menhaden Special Select, Omega Protein Corporation, Houston, TX), was also fed to trout for a total of 4 diets in the study. Each diet was randomly distributed to 3 tanks for each water source. Daily feeding rates were approximately 2.0% body weight and calculated for each individual tank from feeding charts developed at Snake River Farm and based on mean fish weight and the cumulative tank biomass. Beginning at 8 am each day, trout in every tank were fed continuously over a 12 h period with an automatic, clockwork belt feeder (Dynamic Aqua-Supply, Ltd., Surrey, British Columbia, Canada). Rainbow trout were acclimated for 14 days prior to the feeding of experimental feeds, which lasted 12 weeks. After the acclimation phase, trout in each tank were counted and group-weighed, and this process was repeated every 3 weeks for determination of weight gain (g/fish and % gain), apparent feed conversion ratio (FCR), and relative percent survival (RPS), which were calculated and averaged for each dietary treatment and water source combination, and adjust feeding rates for each tank. We use the terms “apparent FCR” and FF (rather than feed intake) since feed charts were utilized that had been optimized for use at the Snake River Hatchery with their production stock of rainbow trout. Daily rations were weighed and recorded prior to loading automatic feeders, which may not be as accurate as weighing feed bottles before and after feeding to determine the amount of actual feed consumed.

Formulae:% Gain = [(g Final weight − g Initial weight)/g Initial weight] × 100
FCR = feed intake (dry weight)/body weight gain (wet weight).
RPS (%) = [1 − (final fish count/initial fish count)] × 100

All experimental diets were produced with a twin-screw, cooking extruder (DNDL-44; Buhler AG, Uzwil, Switzerland) to create extruded, floating pellets as previously described [37]. The final moisture content of the pellets was reduced to below 10%. Experimental feeds were top-coated with dietary oil and allowed to cool before storage in plastic-lined paper bags at room temperature. All diets were fed within 4 months of manufacture.

### 2.3. Fish Sampling and Physiological Analyses

At the conclusion of the study, eight fasting fish (fasted for 48 h) were randomly sampled from each tank and euthanized using a solution of 200 mg/L MS-222. Four were immediately stored on ice and later frozen (−10 °C) for use in proximate analysis. The remaining four fish were used in blood and tissue (gill and liver) sampling for cortisol measurement and stress gene expression analysis, respectively (explained in greater detail below). For proximate analysis, trout whole-body samples were processed into a puree with a Robot Coupe food processor (Robot Coupe R-2, Ridgefield, MS, USA). Duplicate whole-body subsamples were used in proximate analysis. Additionally, three subsamples of each experimental diet were collected for proximate composition analysis. Proximate composition analysis for both feed and fish samples was conducted according to AOAC [45] methods, except for crude protein and crude lipid. The dried samples were ground into a uniform consistency using a mortar and pestle, and crude protein was measured using a LECO nitrogen analyzer (TruSpec N, LECO Corporation, St. Joseph, MI, USA) by multiplying the total nitrogen content by a factor of 6.25. Analysis of crude fat was performed with an Ankom HCl Hydrolysis system (Ankom, Inc., Macedon, NY, USA), while ash content was determined by incinerating samples at 550 °C in a muffle furnace. The energy content of the samples was determined using a Parr bomb calorimeter (Parr Instrument Co., Moline, IL, USA).

For gene expression analysis, gill and liver were chosen as the target tissues. Gills of fish are the primary tissue involved in osmoregulation and are affected by changes in water quality, especially alterations in dissolved oxygen, and stress is known to disrupt metabolic processes and cause oxidative damage in liver [46]. Both tissues are easily accessible for quick removal and storage to prevent RNA degradation. Tissue samples were quickly removed, placed in 2.0 mL cryovials, and snap-frozen in liquid nitrogen immediately after sampling and later stored at −80 °C until analyzed. Whole blood was collected from the caudal vasculature using heparinized (1000 U/mL) tuberculin syringes, dispensed into a 2.0 microcentrifuge tube, and placed on ice. After sampling was complete, whole blood was centrifuged at 5000× *g* for 10 min, and the plasma, along with the tissue samples, was stored at −80 °C until assayed.

Plasma cortisol levels were determined by high-performance liquid chromatography–mass spectrometry (HPLC–MS) using the technique outlined by Turpeinen and Hämäläinen [47] and modified slightly for ion trap measurement [36].

The effect of stress due to diet and water quality was also tested by examining the expression of genes affected by stress selected from a data set compiled by Sanchez et al. [39]. The genes in liver and gill evaluated in this study included: FK506, an immunophilin protein that plays a role in immunoregulation and basic cellular processes involving protein folding and trafficking; DIO2, type II iodothyronine deiodinase; REGPS, regulator of G-protein signaling; Cyp1a, cytochrome P450 1aA2 which is involved in the metabolism of xenobiotics in the body; G6PH, Glucose-6-phosphate dehydrogenase which regulates cellular energy and protects against oxidative damage; GADD45a, whose expression is known to increase following stressful growth conditions and treatment with DNA-damaging agents; and IRF-1, interferon regulatory factor 1 that functions to activate transcription of cellular immune responses. β-actin was used for the standard control. RNA isolation, cycling conditions, and analysis were performed as previously described [48]. Accession numbers and probe and primer sequences are shown in Table 2.

### 2.4. Statistical Analysis

Dietary protein source (fishmeal control, C; soybean meal, HiS; soy protein concentrate, SP; fermented soy protein concentrate, fSP) and water source (1st and 3rd use) were fixed effects in this study. Weight gain (% and g/fish), FCR, FF, RPS, whole-body proximate composition, and stress indices (cortisol and gene expression in liver and gill) were analyzed by two-way analysis of variance (ANOVA). Recorded measurements from individual fish were averaged for each tank (experimental unit) prior to statistical analysis. Pairwise comparisons between means for main effects were made using the Holm–Sidak method following a significant F test by examining mean contrasts with the associated confidence interval at P = α/c, where c = number of pairwise comparisons and α = 0.05. The assumptions of equal variances and normality of data sets were evaluated using the Levene test and Shapiro–Wilk test, respectively. For data expressed as a percentage (e.g., RPS), arcsine transformation was applied prior to analysis using an established methodology [49]. All statistical analyses were conducted using the SAS System for Windows Version 8 (SAS Institute, Inc., Cary, NC, USA). Bar graphs for growth and physiological indices were created with SigmaPlot version 14.0 (SPSS, Inc., Chicago, IL, USA). Heatmaps were generated using R statistical programming language (R Foundation for Statistical Computing, Vienna, Austria) and RStudio (Posit Software, Boston, MA, USA). A significance level of α = 0.05 was used for all statistical analyses.

## 3. Results

### 3.1. Water Quality

Water temperature ranged from 13.7 to 14.2 °C with an average of 14.0 ± 0.3 °C and was not significantly affected by diet (*p* = 0.77) or water source (*p* = 0.66) and well within the tolerable range of rainbow trout (Table 3). Water pH was slightly higher for 1st use water (7.81 ± 0.05) in comparison to 3rd use water (7.72 ± 0.04) but did not differ significantly (p = 0.10). The pH exhibited little variation during the study and was also not affected by diet (*p* = 0.06) (Table 3). Dissolved oxygen levels were near 100% saturation for 1st use water inflow and approximately 80% for 3rd use inflow. Levels decreased after passing through tanks and were significantly lower (*p* < 0.001) in 3rd use water (66.1 to 76.0%) compared to 1st use (81.5 to 88.3%) but not affected by diet (*p* = 0.70) (Table 3). Dissolved oxygen averaged 9.31 ± 0.05 and 7.07 ± 0.88 mg/L for 1st use and 3rd use inflow, respectively, decreasing to approximately 7.80 ± 0.13 mg/L (7.37 to 8.02 mg/L) and 5.91 ± 0.11 mg/L (5.85 to 5.96 mg/L) after passing through tanks (Table 3).

Total dissolved solids (TDS) were lower for inflow compared to water leaving the tanks and significantly higher (*p* = 0.003) for 3rd use (476.6 mg/L) compared to 1st use water (452.7 mg/L) (Table 3). The TDS in water of trout fed the fSP diet (502.3 mg/L) was significantly higher (*p* < 0.001) compared to the other diets (440.6 to 457.8 mg/L), which were similar, regardless of the water source. In the current study, the turbidity levels were low but non-significantly (*p* = 0.18) elevated in 3rd use water (1.939 NTU) in comparison to 1st use (0.092 NTU), and values were also higher exiting the tanks compared to water inflow (Table 3). There was some non-significant (*p* = 0.57) variation for water turbidity among the dietary treatments with values higher for the HiS and fSP feeds (2.004 to 2.972 NTU) compared to the C and SP feeds (0.147 to 0.178 NTU). The former two diets produced the highest turbidity in 3rd use water at approximately 3-4 NTU. None of the observed differences were statistically significant.

Unionized ammonia (UIA; NH_3_) in this study was low and ranged from <0.001 to 0.003 mg/L and was unaffected by water source (*p* = 0.16) and diet (*p* = 0.09). Levels of UIA were generally higher for 3rd use compared to 1st use water in this study but exhibited high variation, and values were also higher exiting the tanks compared to the inflow water (Table 3). Levels of UIA were elevated in 1st use water of trout fed the SP and fSP feeds (0.0016 and 0.0019 mg/L) compared to the other two feeds (0.0001 and 0.0007 mg/L), but UIA did not increase appreciably for individual diets in 3rd use water. Overall, we observed ammonia levels that were higher in tanks receiving the SP diet, but the differences were not significant.

### 3.2. Proximate Composition and Growth Performance

Differences in whole-body proximate composition of rainbow trout were observed due to diet but not water source (Table 4). Whole-body energy was significantly lower (*p* = 0.02) in trout fed the HiS diet (6259.9 kcal/kg) when compared to the C (6443.9 kcal/kg) and SP (6446.3 kcal/kg) but not fSP (6369.0 kcal/kg) diets. These differences in whole-body energy values among dietary treatments were likely due to the crude fat content which was significantly higher (*p* = 0.05) for the C and SP diets than for the HiS diet and fSP. Whole-body protein and ash contents were similar (*p* > 0.05) among diets. The experimental diets had comparable proximate composition with no sizeable variation (*p* > 0.05) among values (Table 5).

Weight gain (g gain/fish) was significantly (*p* = 0.02) lower for trout in 3rd use (183.8 g/fish) compared to 1st use (205.7 g/fish) water (Figure 1), but the effect, although lower for 3rd use water, was not significant when expressed as % gain (*p* = 0.41; 159.4% vs. 153.9%) (Figure 2). Weight gain (g gain/fish and % gain) was also lower (*p* < 0.001) for fish fed the HiS diet (165.2 g/fish and 136.4%) compared to the other diets (201.9 to 206.0 g/fish and 158.4 to 167.2%), which produced similar growth, and the same pattern for weight gain for diet was observed in both 1st and 3rd use water. Apparent FCR followed a similar pattern as weight gain and was significantly higher (worse) and for HiS (1.14) compared to the other diets (0.90 to 0.93) and for 3rd use (1.01) when compared to 1st use (0.94) water (*p* = 0.004; Figure 3). Survival (RPS) was significantly affected by water source (*p* = 0.008) and diet (*p* = 0.05). Survival was lower for trout in 3rd use (91.9%) compared to 1st use (97.6%) water, and RBT fed HiS had lower RPS (90.9%) than for fSP feed (98.3%). The survival of RBT fed the C (94.0%) and SP (95.6%) diets was similar (Figure 4). These dietary trends in survival were similar for trout in 1st and 3rd use water. Significant interactions (*p* > 0.05) were not observed between water source and diet for any proximate or growth parameters.

### 3.3. Physiological and Stress Gene Responses

Plasma cortisol concentrations were low (<10 ng/mL) for rainbow trout regardless of treatment (Figure 5). Neither water source (*p* = 0.82) nor diet (*p* = 0.08) significantly affected plasma cortisol concentrations. There was significant variation in plasma cortisol values, and this variation likely prevented identification of any significant differences among the experimental treatment combinations. The highest cortisol concentrations were observed in rainbow trout fed the SP (7.02 ng/mL) and fSP (5.63 ng/mL) diets with cortisol for trout on the C (4.39 ng/mL) and HiS (3.37 ng/mL) feeds more similar. Less variation in plasma cortisol was observed between diets in 1st use water. Hematocrit was significantly affected by diet (*p* < 0.001) but not water source (*p* = 0.97) (Figure 6). Trout fed the HiS feed had significantly lower HCT compared to the other diets, which were similar to one another. There was a significant interaction between water source and diet for HCT (*p* = 0.02), likely caused by the significantly lower HCT in trout fed HiS across water sources.

In this study, water source had a significant impact on the expression of most genes in gill and liver (Table 5). The exceptions were IRF-1 and Dio2 in liver and FK506bp2 and GADD45A in gill. When there was a significant effect of water source, gene expression typically was upregulated in gill (Figure 7) and downregulated in liver (Figure 8). Diet had less of an effect on gene expression, only affecting REGPS and FK506bp2 in liver and REGPS, Cyp1a, and GADD45A in gill (Table 5). In liver, FK506bp2 and REGPS show similar patterns of expression with trout fed the HiS exhibiting the lowest expression, and SP and fSP showing the highest rates of expression, respectively (Figure 8). Identical patterns of gene expression were observed in gill for REGPS, Cyp1a, and GADD45A with trout fed the C and SP diets exhibiting the highest and lowest expression, respectively, among the diets (Figure 7). Expression in gill for trout on the fSP and HiS were similar to the other diets for these genes. Expression profiles for each gene are also provided as bar graphs for gill (Appendix A) and liver (Appendix A).

## 4. Discussion

### 4.1. Water Quality

In this study, all measured water quality parameters (temperature, TDS, UIA, pH, and turbidity) increased, except for DO which decreased, after passing through tanks (inflow vs. outflow). Water quality was also similarly reduced in tank outflow between 3rd use and 1st use water, which correlated with worse growth performance within an individual diet (e.g., comparing the HiS diet between 1st and 3rd use water). Diet also had a significant impact on growth. The effects of water source and diet on growth will be explored in greater detail below.

Water temperature (and pH) showed little variation during the study. Water turnover in the tanks was three times per hour, similar to raceways at the farm, keeping water temperature fairly constant during the study, and this same pattern is also characteristic for raceway water temperatures (unpublished data), even during the warmer summer months. The study was conducted from mid-September to mid-December, and water temperature was well within the tolerable range for rainbow trout [50].

Dissolved oxygen ranged from approximately 66% to 88% saturation (5.85 mg/L to 8.02 mg/L), with the highest and lowest levels in 1st use and 3rd use water, respectively, with no effect from diet. These levels align closely with DO measurements recorded in comparable raceway systems. A study conducted by Maillard et al. [50] investigated water quality across three farms in West Virginia with DO concentrations ranging from 8.2 to 14.2 mg/L upon entry and 3.2 to 13.3 mg/L upon exit from raceways with the values varying based on the specific farm. Generally, a minimum of 5–6 mg/L DO is required to avoid sublethal, hypoxic effects in farmed trout [30]. In this study, average DO levels in 3rd use water were below 6 mg/L with periodic or instanced DO levels that dropped to around 5 mg/L or slightly lower during 24 h monitoring. These low values predominantly occurred directly after feeding at 9 am and 7 pm and approached the minimum DO limits for rainbow trout. With DO concentrations near the hypoxic threshold for rainbow trout, it is reasonable to conclude that the lower growth observed in 3rd use water was at least in part impacted by these lower DO levels.

Total dissolved solids (TDS) are comprised primarily of inorganic salts, predominantly magnesium, potassium, calcium, sodium, bicarbonates, chlorides, and sulfates with small quantities of organic matter [31]. Toxicity of TDS occurs through increases in salinity, changes in the ionic composition of the water, and toxicity of individual ions. The potential for TDS to adversely affect aquatic organisms is determined by the composition of these dissolved substances [50]. It is difficult to determine the negative effects of elevated TDS because it does not differentiate between ions and other dissolved materials. Although TDS was significantly higher for the fSP diet, it did not appear to have negatively affected growth performance or survival during the 12-week study. However, the high levels of TDS we observed in 3rd use water may negatively impact trout performance over a longer period of time [30]. Furthermore, we have found that soy diets, including those with high levels of SPC, can lead to reduced fecal stability in rainbow trout [25,26] and an increase in effluent nutrient load, in particular phosphorus [51], which can have negative health and environmental impacts.

Diets rich in soy protein have the potential to cause pathomorphological alterations in the distal intestinal epithelium of trout to produce diarrhea-like conditions [32], comprised of fine fecal particles that remain suspended in the quiescent zone of raceways with elevated levels of total suspended solids (TSS) [52,53]. As mentioned, we have shown previously that SBM and SPC can result in higher levels of suspended fine fecal particles compared to a standard FM-based diet [25,26]. The presence of suspended solids can negatively affect fish growth and survival. Although turbidity is not a direct measurement of TSS, a positive, linear correlation often exists between the two [54]. Turbidity was non-significantly elevated in 3rd use water compared to 1st use. Average turbidity was noticeably higher for the SP and fSP diets, but high variation among the turbidity values likely prevented statistically significant differences between diets and water sources. Duchrow and Everhart [54] reported that turbidity measurements can offer valuable insights when a significant portion of the overall turbidity originates from settleable solids, which is generally the case for most hatchery effluent [30]. Lloyd [55] proposed that elevated levels of suspended solids may prove fatal to salmonids, but lower levels of suspended solids and turbidity may lead to chronic sublethal effects, such as diminished foraging capability, compromised growth, susceptibility to disease, and increased stress. Salmonid populations that are unaccustomed to elevated natural turbidity levels or exposure to human-caused sediment sources may be adversely impacted by turbidity levels considered to be low [56]. Turbidity observed in this study ranged from 0 NTU for 1st use inflow to about 3–4 NTU for 3rd use water for the HiS and fSP diets and was likely too low to bring about a negative effect on rainbow trout performance from TSS or turbidity, regardless of water source and diet combination. It should be noted that very large spikes in turbidity (>100 NTU) were periodically observed and associated with fish grading and cleaning in raceways—this undoubtedly caused the high variation among turbidity measurements and likely prevented the detection of differences among treatments. This illustrates that the daily average or single timepoint measurement may not always be the best indicator of water quality, since episodic, acute spikes in water quality parameters could have negative impacts on fish health.

Fish excrete nitrogen as ammonia, which reacts with water to form ammonium ions (NH_4_^+^) in equilibrium with unionized ammonia (NH_3_; UIA) that is dependent upon pH and temperature [30]. Unionized ammonia is highly toxic to fish [57]. Unionized ammonia concentrations were overall low for diets and water sources. Levels were generally higher for 3rd use compared to 1st use water in this study, especially when examining changes for individual diets. Médale et al. [11] reported that ammonia and urea secretion of RBT increased as SPC increased in the diet. We observed higher ammonia levels in tanks receiving the SP and fSP diets, but the differences were not significant. Trout are one of the most sensitive fish species to UIA [58]. The levels of UIA that negatively affect rainbow trout are variable and dependent upon environmental factors (e.g., DO and CO_2_ concentrations, temperature, concentration of other ions, prior exposure), duration of exposure, and stage of growth [58,59], with lethal concentrations ranging from 0.06 to 1.1 mg/L UIA [58,60]. Recommendations for UIA concentrations to avoid chronic effects vary by a factor of three, ranging from 0.0125 to 0.04 mg/L [58,59,60,61,62,63]. However, there is also evidence that much lower UIA levels of 0.004 mg/L can cause gill damage to Chinook salmon (*Oncorhynchus tshawytscha* Walbaum) [59]. Unionized ammonia levels in this study were well below levels (0.001–0.003 mg/L) deemed detrimental to rainbow trout. Although UIA values were higher for 3rd use compared to 1st use water and the SP and fSP diets, high variability among values likely prevented the detection of statistically significant differences, and it is unlikely that trout performance was affected by UIA levels during this 3-month study but may contribute to chronic effects over longer periods.

### 4.2. Growth Performance

Fishmeal is a highly digestible, complete protein source that continues to be used as the primary protein source in rainbow trout feeds [1]. However, it is a finite resource and continues to increase in price [64], which will continue to raise the cost of trout feed in the future. Therefore, trout producers and feed manufacturers continue to search for low-cost, alternative sustainable protein sources for FM. Soybean meal has been the most widely used alternative protein source successfully incorporated into the diets of rainbow trout and other fish species [1,4]. However, rainbow trout and other carnivorous species have a limited tolerance to dietary SBM. In this study, the growth performance (gain and FCR) of rainbow trout in 3rd use water was reduced compared to those in 1st use water. Growth performance was also lower for fish fed the HiS diet compared to the other diets, which were similar. Soybean meal is known to cause pathophysiological issues in rainbow trout when fed in excess, and this can lead to reduced growth and decreases in other performance parameters.

Soybean meal has high levels of structural fiber and anti-nutritional factors (ANF) that limit its incorporation into the diets of rainbow trout [65]. As Hardy [8] reported, inclusion rates of SBM in rainbow trout diets are typically kept at less than 20%, and dietary levels greater than 20% can cause reduced weight gain and feed efficiency. Adelizi et al. [17] found that rainbow trout (35 g, average weight) fed an all-plant diet containing 43% SBM and 22.7% corn gluten meal had significantly higher growth than the other plant-based diets tested, but the growth was lower than for the commercial FM-based feed. Similarly, Dabrowski et al. [66] reported that while the growth performance of rainbow trout fed a diet containing 13% SBM was similar to the FM control, fish fed a 25% SBM diet had significantly lower growth, and those fed 50% SBM exhibited growth arrestment and high mortality. The HiS diet contained approximately 41% defatted SBM with lesser amounts of corn protein concentrate (CPC), wheat gluten meal, and wheat flour and was formulated to meet all nutritional requirements of juvenile RBT. The dietary level of SBM used in this study significantly impaired RBT growth, likely due to the presence of the ANFs mentioned (e.g., phytic acid, protease inhibitors, saponins, lectins, and phytoestrogens). Likewise, FCR for rainbow trout fed the HiS feed was significantly poorer compared to the other feeds.

Modification of soy protein products by biological (e.g., fermentation), chemical (e.g., acid precipitation), and mechanical (e.g., alcohol and moist heat–water leaching) methods can improve its value as a protein source in fish by removal of some fiber, other anti-nutrients, and fractions of low nutritive value to produce high-protein soy concentrates [24,65,67,68]. Researchers have shown that refinement of soy to SPC results in improved feed conversion (and growth) compared to traditional SBM in RBT [69,70], with these improvements caused by reduced ANFs and higher feed intake [18,71]. In this project, the growth of RBT fed diets containing SP and fSP providing 50% of crude protein was equal to the C diet and significantly better than HiS. Similarly, Mambrini et al. [72] reported that growth was not impaired in juvenile RBT in which 50% of FM was replaced with SPC. However, they found that levels greater than 50% reduced growth performance. Other studies support the addition of higher levels of SPC in RBT diets. Kaushik et al. [16] found that the inclusion of SPC in RBT diets at a rate of 22–62% of diet (46% crude protein) had no effect on growth performance. Advances in fermentation of SPC have allowed dietary levels as high as 80% [67] and 100% [68] in the diet as substitution of FM (with lysine and methionine supplementation) to not affect growth performance. However, the level of ammonia and urea excretion increased as SPC increased in the diet in their study. We also observed non-significant increases in ammonia in the water of trout fed the SP or fSP diets. In past research, we determined the negative effects of SPC on fecal stability with reduced water quality—although many ANFs are removed during the process of SBM to SPC, the structural fiber content is little affected [26], which may have contributed to poorer water quality in trout fed these diets.

Survival (RPS) was significantly lower for trout in 3rd use compared to 1st use water and when fed the HiS in 3rd use but not 1st use water. Trout fed the HiS diet in this study were generally in poorer physiological condition, with lower body fat stores and lower growth rates. High levels of SBM in diets of RBT can cause reduced growth, and at levels ≥50% of diets, significant mortality can occur [66]. Water quality was generally poorer for 3rd use water in this study and may have negatively impacted survival. As mentioned, diets high in soy and other plant proteins can induce diarrhea-like symptoms in rainbow trout, causing an increase in TSS, ammonia, and overall reduction in water quality [51,73]. There may have been a combinatorial effect of 3rd use water and HiS on RPS, but the influence of a single water quality parameter on survival was not observed. In this study, the use of SP or fSP to supply 50% of CP in diet did not negatively affect survival or growth. However, due to high variability in some water quality parameters and elevated TDS, further examination of the use of these products on water quality is needed.

### 4.3. Physiological and Stress Gene Responses

Culture conditions and practices are often sources of stress in rainbow trout that lead to physiological responses, which are often initially beneficial but frequently lead to detrimental effects on fish health and well-being if the response is robust or prolonged [74]. Various measures have been used to determine stress homeostasis in fish, including plasma cortisol and blood hematocrit [75]. Hematocrit values did not differ between trout in 1st and 3rd use water and do not suggest the presence of an active stress response. However, HCT was lower in trout fed the HiS feed compared to the other diets. Low HCT can indicate that organisms, including fish, are experiencing a stress response or are in poor physiological condition [76]. Trout fed the HiS diet had poorer growth and survival, indicating that they were in poorer physical condition compared to the other dietary treatments, and this supports the proposed hypothesis for low HCT values that these fish were in a poor physical state. 

Plasma cortisol concentrations for rainbow trout were low and <10 ng/mL regardless of treatment or treatment combination of water source and diet. By comparison, Barton [34] previously documented pre-stress and post-stress cortisol levels (after netting and 30 s aerial exposure) at 1.7 and 43 ng/mL in plasma, respectively, which indicates that the trout in our study were not undergoing acute physiological stress when sampled at the end of the project. In contrast, rainbow trout exposed to reused water marked by elevated carbon dioxide levels and low dissolved oxygen display significantly higher cortisol concentrations (105 ng/mL) than the trout in our study [39]. The difference in cortisol response levels may be attributed to the acute nature of their exposure, spanning only a few hours without sufficient time for adaptation—unlike the current study, where fish were sampled 12 weeks after stocking. Exposing rainbow trout to municipal wastewater effluent for a period of 14 days causes chronic stress and disrupts their adaptive response [77]. Despite the comparatively low plasma cortisol concentrations in this study, the gene expression data suggest that the rainbow trout exposed to 3rd use water displayed subcellular alterations attributable to exposure to reduced water quality. This indicates that the trout in 3rd use water were likely undergoing chronic stress.

Previous research has shown that stress has an appreciable effect on gene expression [39,40,78]. However, little information exists on the expression of genes in rainbow trout exposed to hatchery effluent of varying water quality. Water source exhibited a significant impact on the expression of most genes in gill and liver in this study (except for IRF-1 and Dio2 in liver and FK506bp2 and GADD45A in gill) in which gene expression was typically downregulated in liver and upregulated in gill. In water, the gills are the first organ to come into contact and react in response to water quality changes. It has been shown that increased stress causes several changes in cell regulatory pathways [40,78] with most gene responses as an increase in expression as the cells initially cope with the stress, which is what we observed in the gill for most of the genes evaluated in this study. Gagne et al. [35] found that the expression of several genes (metallothionein, cytochrome P450 3a4, and vitellogenin) in liver of trout showed a significant negative correlation (increased mRNA levels) to flow rate, TSS, alkalinity, nitrogen, and DO, but expression of superoxide dismutase (SOD), glutathione S-transferase, and multidrug resistance transporter was largely unaffected. These genes were not evaluated in the present study. However, Cyp1a, which showed significant downregulation in liver of trout in 3rd use water in our study, was unaffected by measured water quality parameters in Gagne et al. [35]. The liver has some immune response capabilities, but its major functional capacity is as a metabolic organ processing incoming nutrients. In most stress situations, animals respond directly but over time to reduce metabolic activity, this mirrors the downregulation we see for the majority of the genes in liver in this study, which is similar to what Du et al. [79] observed in liver of *Coilia nasus*.

In a previously published study [36], we evaluated the effect of trout strain and water source (1st, 3rd, and 5th use water) on expression of a slightly different set of stress-related genes (CAT, FKBP2, SOD, GPX1, FKBP2, GADD45a, and REGPS; see Welker et al. [36] for descriptions) in liver, gill, kidney, and spleen with only the latter two genes analyzed in the present study. We found that all the genes examined were responsive to changes in water quality between raceway sources, but the greatest changes in expression occurred in trout in 5th use water. The change in expression of REGPS and GADD45A in gill and liver were similar in trout between 1st and 3rd use water between the two studies except for the expression of liver GADD45A, which was unchanged in our previous study. In our previous work, expression of SOD, GPX1, and CAT, associated with the antioxidant enzymes superoxide dismutase, glutathione peroxidase, and catalase, which function in the reduction of specific reactive oxygen species (ROS) [80], was altered between 1st, 3rd, and 5th use water. We observed upregulation of CAT and GPX1 and downregulation of SOD in response to 5th use water exposure, suggesting that the associated water quality stress may have upset the antioxidant–oxidant balance leading to oxidative stress. Research has shown that the physiological stress response can upset the balance between ROS, by-products of normal cellular metabolism, and their antagonistic antioxidant counterparts, causing oxidative stress [81,82] which can damage cell components, such as lipids, proteins, and DNA in mammals [83] and fish [84,85]. When oxidative stress occurs, the expression of genes associated with antioxidant enzymes and the repair of damaged cellular components are altered, often causing an increase in the synthesis of these enzymes and their biological activities [80]. Research has also linked the mammalian forkhead proteins and growth arrest and DNA-damage-inducible proteins, such as GADD45A, to increased stress resistance. Research has demonstrated that in mammalian cells, stresses, including exposure to the ROS hydrogen peroxide, induce transcription of GADD45A [86]. In this project, GADD45A was either unchanged (gill) between 1st and 3rd use water or downregulated (liver). However, we found that exposure to further degraded water quality in 5th use water induces increased expression of GADD45A [36], suggesting that the water quality stress was not sufficient to affect this gene (and other evaluated genes) in 3rd use water compared to 5th use.

The effect of dietary protein source on gene expression was less than water source and only affected REGPS, Cyp1a, and GADD45A in gill and REGPS and FK506bp2 in liver. Trout fed the HiS diet showed downregulation of FK506bp2 and REGPS in liver, while those fed SP and fSP exhibited upregulation of these two genes. Gill REGPS, Cyp1a, and GADD45A were upregulated in trout fed the C diet and downregulated when given the SP feed. Abernathy et al. [87] examined global gene expression in muscle and liver of rainbow trout strains selected and non-selected for soy tolerance and fed a high-soy diet. Expression patterns differed between the selected and non-selected strains, and although none of the genes examined in the present study made their candidate gene list (based on fold change in expression between trout strains), several genes involved in oxidative stress homeostasis showed significant changes in expression, as observed in this study and our prior research. Acute and chronic immune responses are probably responsible for some of these relative changes associated with a high SBM diet, as noted in Abernathy et al. [87].

These results demonstrate that the diet and dietary components, such as soy, affect the expression of stress genes in fish. The effect can be positive, as is shown with the HiS diet, where changes in water quality had the least overall effect on stress, or negative, where the fSP had an overall higher expression level for stress genes between diets in the sample water use. This study and our previous research also show that exposure to hatchery effluent influences the expression of stress-related genes and the oxidative stress response in rainbow trout with 5th use water eliciting a more pronounced gene response than 3rd use water. In addition, other physiological systems and whole-body responses, such as growth, can be affected by long-term exposure to stress [75]. For example, chronic stress is known to suppress the growth of rainbow trout from alterations in protein synthesis [88]. Future evaluations regarding dietary and water quality stress effects should consider both acute and chronic stress responses as they affect these system-wide responses.

## 5. Conclusions

Soy protein and water sources had significant effects on the growth and survival of RBT. Water quality was generally poorer for 3rd use water and negatively affected growth and survival. Trout fed the HiS diet had reduced growth and lower survival in 3rd use water. While this may have been a combinatorial effect, the influence of HiS on individual, measured water quality parameters was not detected. Rainbow trout fed diets containing the SP or fSP products providing 50% of dietary protein had similar growth to the C diet and significantly better growth and survival compared to trout fed HiS. The effect of protein source on water quality was difficult to determine. The water of trout fed the SP diet showed significantly higher TDS. Other water quality parameters were also higher for the soy protein sources in some instances, but high variability among values made the determination of effects inconclusive. Production operations in commercial, flow-through hatcheries produce greater variability in rearing conditions compared to laboratory tank systems supplied with high-quality, well or spring water, which likely produces greater variability in measured water and physiological parameters and makes detection of biologically significant effects challenging. Although not indicated by cortisol values, differential expression of several stress-related genes in this and our previous work indicates that stress effects on a subcellular level are present that likely affect the physiological well-being of rainbow trout in relation to diet and water quality, which may be interrelated. Due to a projected increase in fish production for human consumption, sustainable protein sources will be necessary to replace FM, and further incorporation of plant protein sources, including soy, will be needed. However, additional evaluation is required to balance dietary incorporation and impacts on water quality. The ability to correlate the expression response of immune genes with water and diet quality may be an invaluable tool for the evaluation of feed formulations and fish stocks under commercial conditions.

## Figures and Tables

**Figure 1 animals-13-03090-f001:**
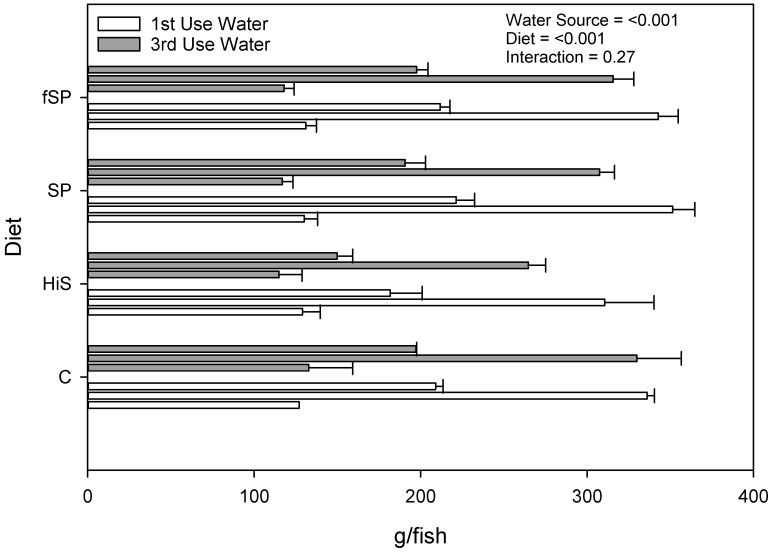
Growth (mean ± S.D.) of rainbow trout fed the four experimental feeds. For each water use group (1st and 3rd), initial weight (bottom bar), final weight (middle bar), and weight gain (top bar) are shown. *P*-values indicate the significance of effect of experimental treatments on weight gain (g/fish). S.D. = standard deviation; C = fishmeal control; HiS = high soybean meal; SP = soy protein concentrate; fSP = fermented soy protein concentrate.

**Figure 2 animals-13-03090-f002:**
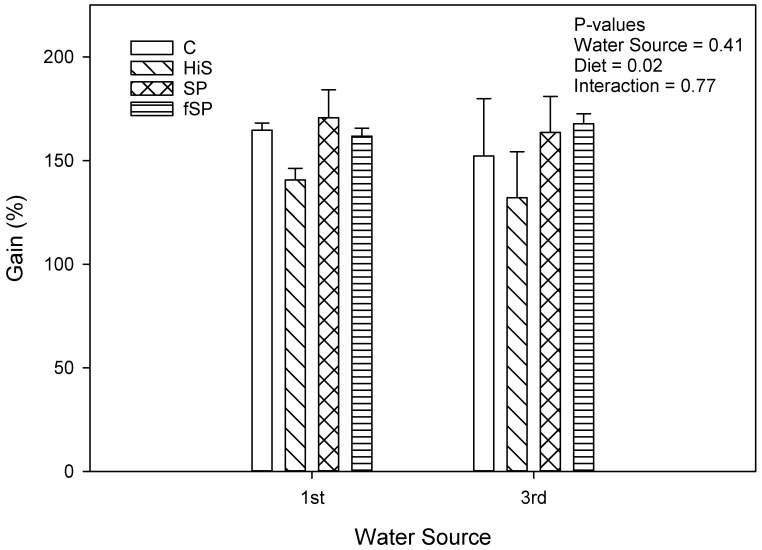
Growth (% gain) (mean ± S.D.) for rainbow trout fed the four experimental feeds reared in 1st use or 3rd use water. S.D. = standard deviation; C = fishmeal control; HiS = high soybean meal; SP = soy protein concentrate; fSP = fermented soy protein concentrate.

**Figure 3 animals-13-03090-f003:**
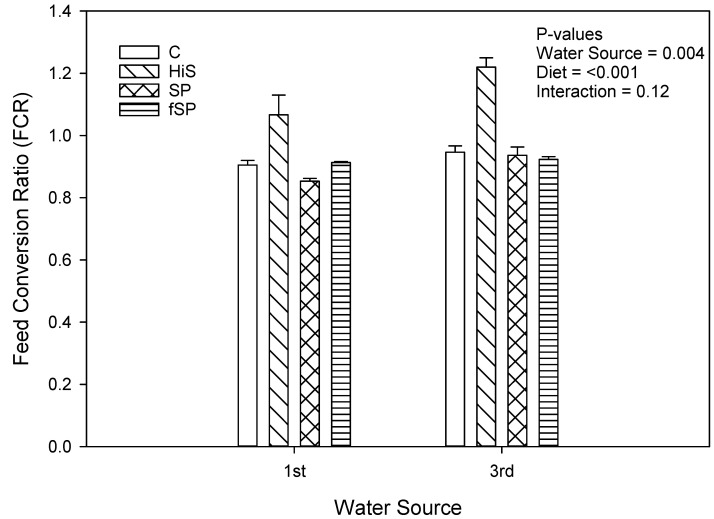
Feed conversion ratio (mean ± S.D.) for rainbow trout fed the four experimental feeds reared in 1st use or 3rd use water. S.D. = standard deviation; C = fishmeal control; HiS = high soybean meal; SP = soy protein concentrate; fSP = fermented soy protein concentrate.

**Figure 4 animals-13-03090-f004:**
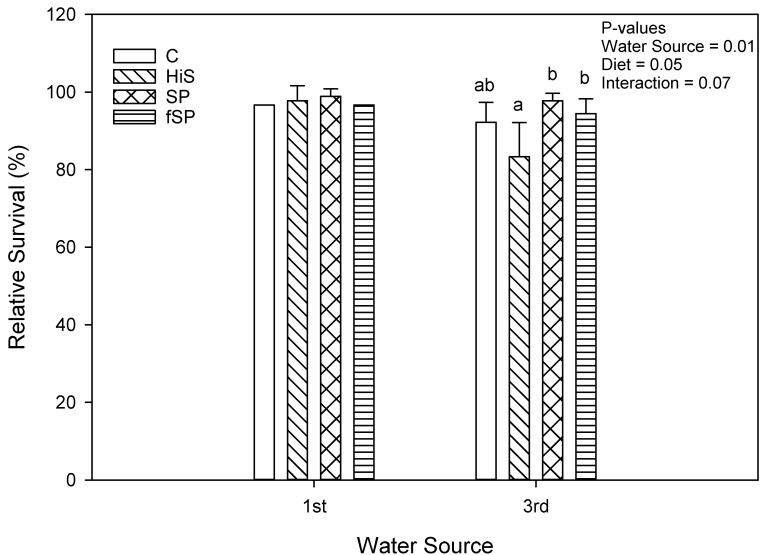
Relative percent survival (RPS) (mean ± S.D.) for rainbow trout fed the four experimental feeds reared in 1st use or 3rd use water. S.D. = standard deviation; C = fishmeal control; HiS = high soybean meal; SP = soy protein concentrate; fSP = fermented soy protein concentrate. Relative survival values denoted by different letters (a, b) are statistically different (*p* < 0.05).

**Figure 5 animals-13-03090-f005:**
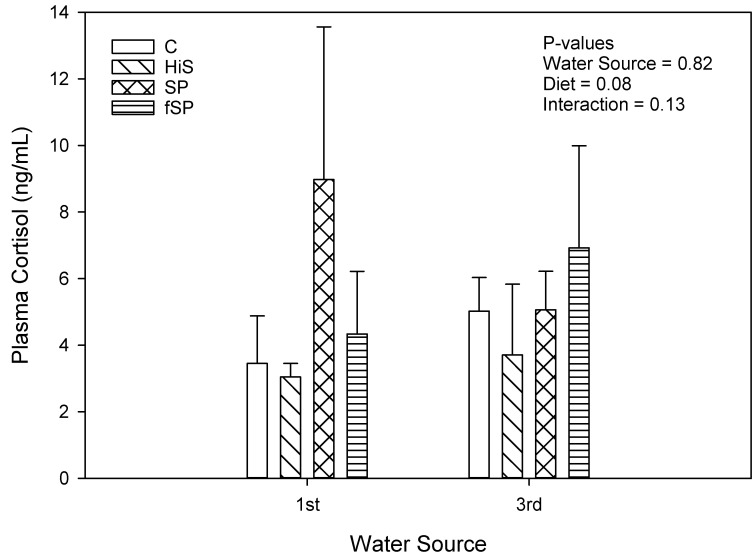
Plasma cortisol concentrations (mean ± S.D.) for rainbow trout fed the four experimental feeds reared in 1st use or 3rd use water. S.D. = standard deviation; C = fishmeal control; HiS = soybean meal; SP = high soy protein concentrate; fSP = fermented soy protein concentrate.

**Figure 6 animals-13-03090-f006:**
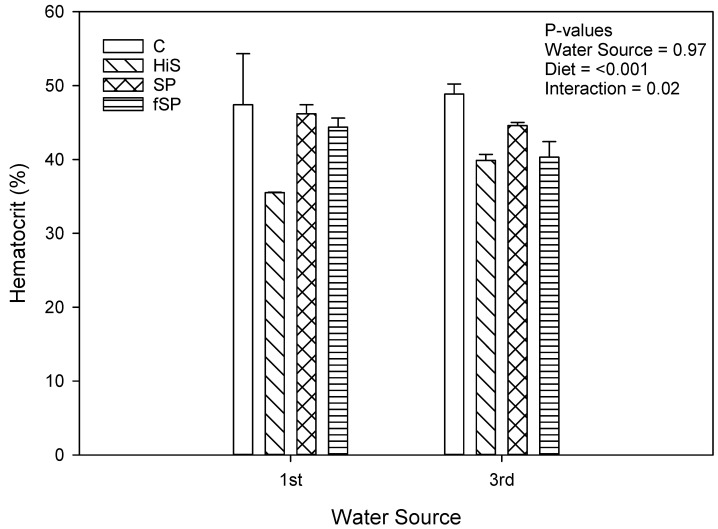
Hematocrit values (mean ± S.D.) for rainbow trout fed the four experimental feeds reared in 1st use or 3rd use water. S.D. = standard deviation; C = fishmeal control; HiS = high soybean meal; SP = soy protein concentrate; fSP = fermented soy protein concentrate.

**Figure 7 animals-13-03090-f007:**
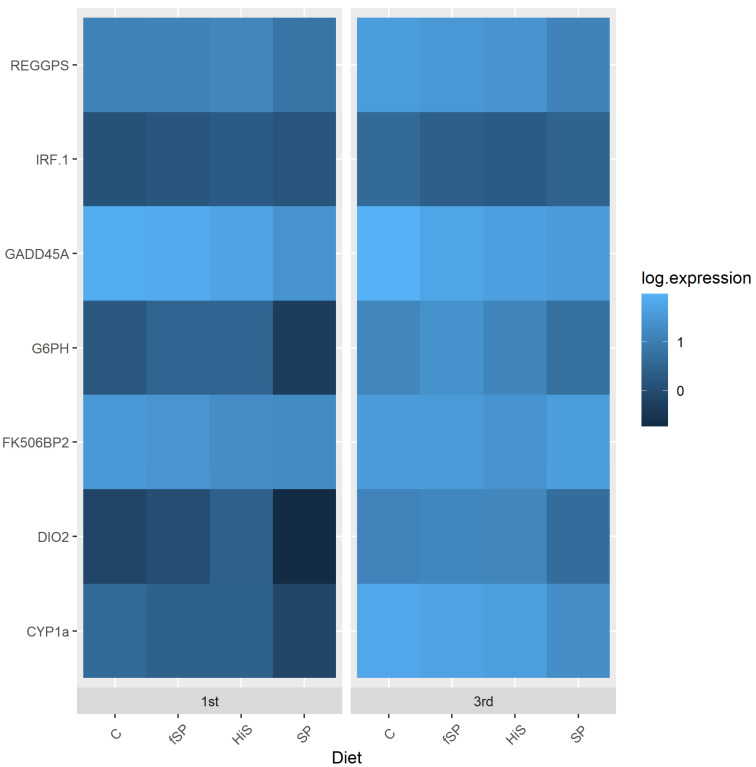
Expression (log 10) heatmap for stress-related genes in gill of rainbow trout fed different experimental feeds and reared in 1st use (1st) or 3rd use (3rd) water. C = fishmeal control; fSP = fermented soy protein concentrate; HiS = high soybean meal; SP = soy protein concentrate.

**Figure 8 animals-13-03090-f008:**
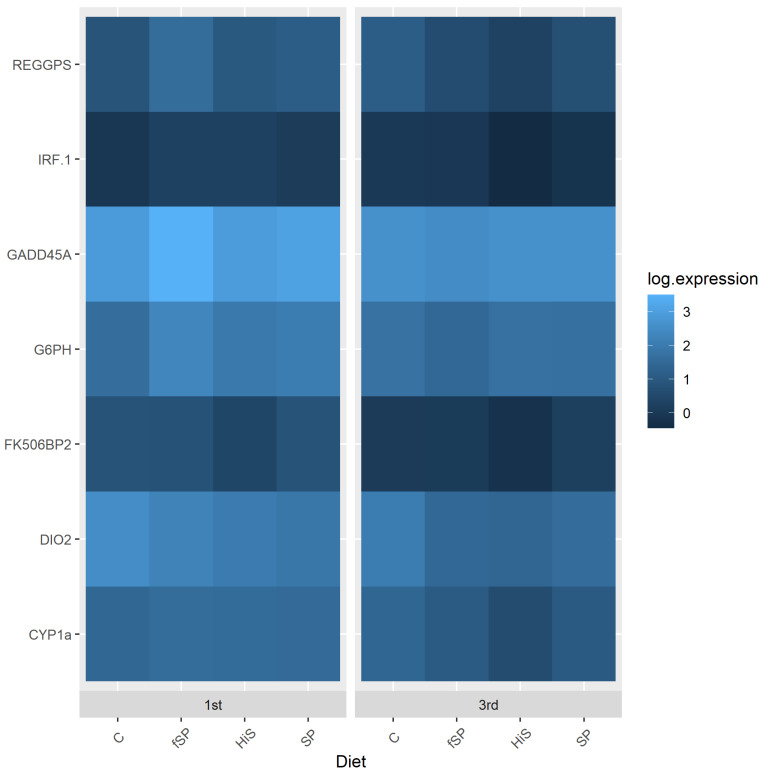
Expression (log 10) heatmap for stress-related genes in liver of rainbow trout fed different experimental feeds and reared in 1st use (1st) or 3rd use (3rd) water. C = fishmeal control; fSP = fermented soy protein concentrate; HiS = high soybean meal; SP = soy protein concentrate.

**Table 1 animals-13-03090-t001:** Ingredient composition of experimental diets for experiment in serial reuse system. C = fishmeal control; HiS = high soybean meal; SP = soy protein concentrate; fSP = fermented soy protein concentrate.

Grams/100 g	C	SP	HiS	fSP
Fish meal ^a^	23.00	--	--	--
Soy protein concentrate ^b^	--	24.64	--	--
Soybean meal ^c^	13.38	13.30	41.25	--
Soy protein concentrate ^d^	--	--	--	30.00
Poultry meal ^e^	13.80	--	--	25.00
Corn protein concentrate ^f^	5.53	17.54	14.65	--
Blood meal ^g^	3.20	--	--	--
Wheat gluten meal ^h^	--	--	6.00	--
Wheat flour ^i^	20.30	16.45	9.87	23.46
Menhaden oil ^j^	15.80	19.49	19.49	14.40
Lysine HCl	1.25	1.79	1.99	0.87
Methionine	0.45	0.55	0.55	0.55
Threonine	0.45	0.22	0.38	0.08
Taurine ^k^		0.50	0.50	0.50
Dicalcium phosphate	---	2.65	2.45	2.25
Vitamin premix ^l^	1.00	1.00	1.00	1.00
Choline chloride ^m^	0.60	0.60	0.60	0.60
Trace min premix ^n^	0.10	0.10	0.10	0.10
Vitamin C ^o^	0.20	0.20	0.20	0.20
Potassium Chloride	0.56	0.56	0.56	0.56
Sodium Chloride	0.28	0.28	0.28	0.28
Magnesium oxide	0.05	0.05	0.05	0.05
Astaxanthin ^p^	0.08	0.08	0.08	0.08
Calculated analysis, %				
Crude Protein, %	40.1	40.2	40.2	40.2
Lipid, %	20.1	20.1	20.0	20.0
Phosphorus, %	0.97	0.94	0.92	0.92
Measured analysis, %				
Crude Protein, %	44.9	43.5	43.0	41.9
Lipid, %	18.8	18.0	18.6	17.7
Energy, kcal/kg	5602.5	5543.3	5646.9	5505.1
Ash, %	8.63	5.90	5.99	6.93

^a^ Menhaden Special Select, Omega Proteins Corp, (Reedville, VA), 610 g/kg crude protein; ^b^ Solae (now known as Bunge, St. Louis, MO), Pro-Fine VF, 693 g/kg crude protein, made by an aqueous ethanol leach method; ^c^ Archer Daniels Midland Company, (Dekalb, IL), 472 g/kg crude protein, made from conventional soybeans; ^d^ Hamlet Protein, (Horsens, Denmark), HP-300, 560 g/kg crude protein, made by an enzymatic treatment of soy meal; ^e^ IDF Inc., (Springfield, MO), 832 g/kg crude protein; ^f^ Cargill, Inc. (Baltimore, MD), Empyreal 75, 756 g/kg crude protein; ^g^ Wilbur-Ellis, (San Francisco, CA), 892 g/kg crude protein; ^h^ Manildra Milling, Inc. (Mission, KS), 778 g/kg crude protein; ^i^ Manildra Milling, Inc., 120 g/kg crude protein; ^j^ Omega Protein, Inc., (Reedville, VA), Virginia Prime menhaden oil; ^k^ NB Group Co. LTD, (Shandong, China); ^l^ DSM Nutritional Products, (Parsippany, NJ), ARS 702: contributed, per kg diet; vitamin A 9650 IU; vitamin D 6600 IU; vitamin E 132 IU; vitamin K3 1.1 gm: thiamin mononitrate 9.1 mg; riboflavin 9.6 mg; pyridoxine hydrochloride 13.7 mg; pantothenate DL-calcium 46.5 mg; cyancobalamin 0.03 mg; nicotinic acid 21.8 mg; biotin 0.34 mg; folic acid 2.5 mg; inostitol 600 mg; ^m^ Balchem Corporation, (Montvale, NJ); ^n^ Sigma Aldrich, (St. Louis, MO). Contributed in mg/kg of diet; manganese 13; iodine 5; copper 9; zinc 40; ^o^ Stay-C, 35%, DSM Nutritional Products; ^p^ Carophyll Pink 10, DSM Nutritional Products.

**Table 2 animals-13-03090-t002:** Gene names, accession numbers and sequences.

	Genebank	Primer Efficiency (%)		
Gene	Accession #	Gill	Liver	Primer/Probe Sequence (Listed 5′-3′)
β-actin	AF254414	99.2	98.3	BactinF: CCCTCTTCCAGCCCTCCTT
				BactinR: AGTTGTAGGTGGTCTCGTGGATA
				BactinMGB: 6FAM-CCGCAAGACTCCATACCGA-NFQ
FK506bp2	NM_001165227	95.6	93.0	FK506bp2OM-F: GAACCAGCCCTTCACATTTACTCT
				FK506bp2OM-R: CTTCTCCCCCTCACACATTCC
				FK506bp2OM-MGB: CTGGTCCCAGCCTTTG
Cyp1a	AF015660	98.9	96.8	CYPF: CAGACTTCATTCCCATCCTTCGTTA
				CYPF: CACAAAGTTGTTGAAACGGTCATTG
				CYPMGB: 6FAM-CCTGCCCAACCGCACC-NFQ
DIO2	NM_001123268	96.1	97.2	DIO2F: CTCCAAAGTGGTCAAGGTTCCT
				DIO2R: CGTGGTGCTGGTCAAGCT
				DIO2MGB: 6FAM-CCGCCGGATGCTACC-NFQ
GADD45a	CA058640.1	101.8	95.0	GADD45aF: CCTCCACAGGGTAATCCAGAAC
				GADD45aR: GCTGTAACCCAGGACTCAATGTG
				GADD45aMGB: 6FAM-CTGCACTGCCATCCC-FQ
G6PH	AF157514	94.2	95.7	G6PHF: CCTCCTCCTCCTCACAAGCT
				G6PHR: CAGGAGAGCACGGTACATGATTTAA
				G6PHMGB: ATGGGTGCTGTGGTCACA
IRF-1	NM_001124293.1	97.3	94.8	IRF1F: GAAGACAGTCACCAAGAAACCCTTA
				EIRF1R: GCTCAGGAACCTCTTGTCGTTT
				IRF1MGB: 6FAM-ACACTGCCTTGCTCCC-NFQ
REGPS	BT074111	97.2	95.0	REGPSF: TCTCATCAGGCGGAATGTGAAG
				REGPSR: CTCTGGGCCTCGTCGAA
				REGPSPR: 6FAM-ACGCCCACCACAGTTT-NFQ

**Table 3 animals-13-03090-t003:** Water quality parameters measured during the study ^1,2^.

Source	Diet	Temp	pH	DO	DO	TDS	NH_3_	Turbidity
		°C		%	mg/L	mg/L	mg/L	NTU
Diet (D)								
	FM-C	14.0 ± 0.7	7.80 ± 0.06	75.4 ± 9.0	6.62 ± 0.15	457.8 ± 31.4b	0.0007 ± 0.0012	0.147 ± 0.227
	SPC	14.0 ± 0.8	7.72 ± 0.06	79.3 ± 11.7	6.90 ± 1.08	502.3 ± 26.0a	0.0022 ± 0.0019	0.178 ± 0.288
	SBM	14.0 ± 0.8	7.78 ± 0.05	74.0 ± 11.5	6.96 ± 0.86	451.4 ± 23.7b	0.0006 ± 0.0016	2.972 ± 7.363
	SPC-f	13.6 ± 0.9	7.76 ± 0.08	73.9 ± 11.4	6.94 ± 1.06	440.6 ± 19.4b	0.0014 ± 0.0013	2.004 ± 2.110
**Water Source (WS)**	1st Use	13.9 ± 0.9	7.78 ± 0.07	85.2 ± 6.0x	7.80 ± 0.13x	452.7 ± 32.7x	0.0011 ± 0.0016	0.092 ± 0.307
	3rd Use	14.0 ± 0.7	7.76 ± 0.07	69.0 ± 7.3y	5.91 ± 0.11y	476.6 ± 33.3y	0.0014 ± 0.0017	1.939 ± 5.006
**D × WS**								
1st Use	Inlet	13.5 ± 0.7	7.84 ± 0.04	99.9 ± 0.5	9.31 ± 0.05	473.1 ± 21.7	0.0004 ± 0.0005	0.000 ± 0.000
	FM-C	13.9 ± 0.9	7.81 ± 0.07	81.5 ± 3.2	7.37 ± 0.30	436.4 ± 22.4	0.0007 ± 0.0008	0.001 ± 0.018
	SPC	13.9 ± 0.9	7.73 ± 0.04	88.3 ± 5.9	7.84 ± 0.54	490.8 ± 17.9	0.0019 ± 0.0024	0.002 ± 0.004
	SBM	13.9 ± 1.1	7.79 ± 0.02	86.0 ± 10.3	7.97 ± 0.95	431.6 ± 12.2	0.0001 ± 0.0000	0.443 ± 0.656
	SPC-f	13.6 ± 1.4	7.83 ± 0.02	85.5 ± 0.1	8.02 ± 0.01	429.9 ± 19.8	0.0016 ± 0.0021	0.000 ± 0.020
3rd Use	Inlet	13.7 ± 0.5	7.74 ± 0.04	76.0 ± 9.5	7.07 ± 0.88	432.0 ± 7.6	0.0010 ± 0.0001	0.248 ± 0.201
	FM-C	14.1 ± 0.7	7.78 ± 0.05	70.3 ± 9.2	5.87 ± 0.85	475.6 ± 27.0	0.0008 ± 0.0016	0.262 ± 0.261
	SPC	14.1 ± 0.8	7.70 ± 0.09	71.1 ± 8.4	5.96 ± 0.77	514.9 ± 29.4	0.0026 ± 0.0013	0.354 ± 0.330
	SBM	14.0 ± 0.7	7.78 ± 0.06	68.1 ± 6.4	5.96 ± 0.59	461.2 ± 22.1	0.0009 ± 0.0020	4.237 ± 8.989
	SPC-f	13.6 ± 0.7	7.71 ± 0.07	66.1 ± 5.7	5.85 ± 0.53	447.7 ± 19.1	0.0014 ± 0.0011	3.339 ± 1.487
**ANOVA**								
Water Source (WS)	F-value	0.20	2.92	41.88	125.70	10.83	2.079	1.877
	*p*-value	0.66	0.10	<0.001	<0.001	0.003	0.161	0.182
Diet (D)	F-value	0.38	2.81	0.48	1.01	13.16	2.418	0.681
	*p*-value	0.77	0.06	0.70	0.41	<0.001	0.088	0.572
Interaction (D × WS)	F-value	0.03	0.97	0.62	0.72	0.33	0.506	0.485
	*p*-value	0.99	0.42	0.61	0.55	0.80	0.682	0.696

^1^ Values are an average of 3 tanks per diet. ^2^ Diets (a, b) or water source (x, y) with different superscript letters are significantly different. Temp = temperature; DO = dissolved oxygen; TDS = total dissolved solids; NH_3_—unionized ammonia; C = fishmeal control; SP = soy protein concentrate; HiS = high soybean meal; fSP = fermented soy protein concentrate.

**Table 4 animals-13-03090-t004:** Proximate composition (%) of whole rainbow trout ^1,2^.

Water Source	Diet	Energy ^3^	Protein	Fat	Ash
1st Use	Control	6496.9 ± 70.3	55.8 ± 1.0	34.8 ± 1.1	7.41 ± 0.04
	SP	6480.6 ± 112.8	56.1 ± 2.3	33.9 ± 3.2	7.45 ± 0.58
	HiS	6291.9 ± 89.1	58.9 ± 1.6	30.6 ± 1.7	7.64 ± 0.54
	fSP	6361.0 ± 68.0	58.5 ± 1.3	30.3 ± 0.4	7.56 ± 0.28
3rd Use	Control	6390.7 ± 56.2	56.8 ± 2.0	32.9 ± 1.4	7.81 ± 0.70
	SP	6412.0 ± 73.6	57.4 ± 0.7	33.1 ± 0.6	6.96 ± 0.35
	HiS	6227.8 ± 184.6	59.0 ± 4.4	30.1 ± 5.2	7.81 ± 0.17
	fSP	6377.0 ± 16.7	57.2 ± 2.5	32.2 ± 1.5	7.09 ± 1.25
***p*-Value**					
Water Source		0.19	0.79	0.73	0.71
Diet		0.02	0.25	0.05	0.47
Interaction		0.76	0.77	0.63	0.56
**Diet Avg.**					
	Control	6443.9 ± 63.2a	56.3 ± 1.5	33.9 ± 1.2a	7.61 ± 0.37
	SP	6446.3 ± 93.2a	56.8 ± 1.5	33.5 ± 1.9a	7.21 ± 0.46
	HiS	6259.9 ± 136.9b	59.0 ± 3.0	30.4 ± 3.4b	7.73 ± 0.36
	fSP	6369.0 ± 42.3ab	57.9 ± 1.9	31.3 ± 1.0ab	7.32 ± 0.77

^1^ Values are an average of 3 tanks per diet (4 fish per tank) expressed on a dry weight basis. ^2^ Diet but not water source nor their interaction had a statistically significant effect on proximate composition. Diets with different superscript letters are significantly different. ^3^ Kcal/kg. C = fishmeal control; SP = soy protein concentrate; HiS = high soybean meal; fSP = fermented soy protein concentrate.

**Table 5 animals-13-03090-t005:** Stress gene expression in liver and gill of rainbow trout reared in 1st and 3rd use water and fed diets with different protein sources.^1.^

		Water Source				Diet					Interaction
Gene	Tissue	1st Use	3rd Use	Expression	*p*-Value	HiS	C	fSP	SP	*p*-Value	*p*-Value
*IRF-1*	Liver	1.201 ± 0.752	0.880 ± 0.365	Unchanged	0.06	0.980 ± 0.710	1.005 ± 0.441	1.157 ± 0.721	0.987 ± 0.475	0.61	0.16
	Gill	1.242 ± 0.362 x	1.597 ± 0.577 y	Up	0.002	1.372 ± 0.403	1.577 ± 0.781	1.363 ± 0.454	1.423 ± 0.391	0.89	0.14
*FK506bp2*	Liver	2.063 ± 1.115 x	1.065 ± 0.386y	Down	<0.001	1.159 ± 0.737 ^a^	1.570 ± 1.056 ^ab^	1.672 ± 1.019 ^b^	1.774 ± 0.943 ^b^	0.01	0.95
	Gill	4.003 ± 1.603	4.702 ± 1.615	Unchanged	0.06	3.940 ± 2.110	4.795 ± 1.501	4.524 ± 1.395	4.283 ± 1.4808	0.08	0.46
*Dio2*	Liver	8.609 ± 7.723	5.257 ± 4.467	Unchanged	0.07	5.809 ± 5.909	9.549 ± 6.745	6.802 ± 7.368	5.728 ± 5.285	0.15	0.82
	Gill	0.990 ± 1.140x	2.919 ± 2.837 y	Up	<0.001	2.381 ± 2.760	2.212 ± 2.557	2.191 ± 2.548	1.238 ± 1.501	0.32	0.61
*REGPS*	Liver	3.327 ± 2.042 x	2.097 ± 1.057 y	Down	<0.001	2.045 ± 1.252 ^a^	2.837 ± 1.142 ^ab^	3.326 ± 2.522 ^b^	2.558 ± 1.299 ^ab^	0.02	0.003
	Gill	2.884 ± 0.932 x	4.182 ± 2.332 y	Up	<0.001	3.709 ± 1.495 ^ab^	4.149 ± 2.620 ^a^	3.789 ± 1.766 ^ab^	2.676 ± 1.431 ^b^	0.003	0.86
*Cyp1a*	Liver	4.588 ± 2.986 x	2.874 ± 1.499 y	Down	0.003	3.281 ± 2.828	4.055 ± 1.898	3.837 ± 2.655	3.664 ± 2.420	0.19	0.14
	Gill	1.419 ± 0.865 x	5.125 ± 2.724 y	Up	<0.001	3.322 ± 2.807 ^ab^	4.374 ± 3.258 ^a^	3.577 ± 2.947 ^ab^	2.309 ± 1.685 ^b^	0.004	0.63
*G6PH*	Liver	7.732 ± 4.245 x	5.281 ± 2.180 y	Down	0.005	6.303 ± 3.431	5.511 ± 1.978	7.196 ± 4.903	6.645 ± 2.991	0.68	0.02
	Gill	1.366 ± 1.7366 x	3.139 ± 3.082 y	Up	<0.001	2.443 ± 2.417	2.463 ± 2.412	2.879 ± 3.682	1.408 ± 1.660	0.17	0.66
*GADD45A*	Liver	23.275 ± 20.460 x	13.169 ± 9.329 y	Down	0.006	16.072 ± 9.539	15.459 ± 14.991	22.393 ± 21.670	17.661 ± 16.924	0.54	0.40
	Gill	5.668 ± 2.364	5.820 ± 2.074	Unchanged	0.68	5.506 ± 1.808 ^ab^	7.046 ± 2.533 ^a^	6.128 ± 2.447 ^ab^	4.514 ± 1.181 ^b^	0.007	0.41

^1^ Diets (a, b) or water source (x, y) with different superscript letters are significantly different. FCR = feed conversion ratio; HCT = hematocrit; C = fishmeal control; SP = soy protein concentrate; HiS = high soybean meal; fSP = fermented soy protein concentrate.

## Data Availability

The data presented in this study are available on request from the corresponding author. The data are not publicly available due to the policy of the funding agency.

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
