# Peer review of "Effect of Dietary Soy Protein Source on Effluent Water Quality and Growth Performance of Rainbow Trout Reared in a Serial Reuse Water System"

_animals, 2023, doi:10.3390/ani13193090_

Round 1
Reviewer 1 Report
This manuscript presents a study to evaluate the effects of different dietary soy protein sources on effluent water quality and growth performance of Rainbow Trout reared in a serial reuse water system. This topic is meaningful. But there are still some things that need to be revised carefully. Overall, I think the manuscript should be reconsidered after revision.
1. In the Abstract and Result section, the description of results with significant differences should be accompanied by P-value.
2. In Table 3, 4 and 6, data with significant differences are not marked with superscript letters. Please check and correct them.
3. All the data in this manuscript are presented in tables. It is suggested that the authors consider changing the results of gene expression analysis into figures to make the data presentation more diversified.
4. the author combined the results and discussion into one section for description. It is recommended that the author divide this section into two parts and describe them separately. Especially in the results section, explain the relationship between the various data in each table. This will make it easier for the reader to understand this article.
Author Response
I have completed my revisions to the manuscript. I thank the reviewers for their suggestions to improve it. The edited manuscript is in the Animals template, but I could not get the tables and figures to align properly when placing them throughout the text, so I included them at the end. The Discussion was edited pretty heavily. Therefore, I copied and pasted a revised version rather than show the track changes. I am open to further improvements if needed. Thank you.

Reviewer 2 Report
I appreciate the authors for trying to test different protein sources at farm scale and evaluate the interaction with water quality after multiple use in a raceway system. Replacing ingredients like fishmeal is utmost research to reduce the environmental impacts in aquaculture.
As far as I am concerned, the manuscript is well structured and the results were presented clearly. I would suggest the authors to consider some points discussed below and revise the manuscript accordingly.
First, I consider the M&M section too long and full of technical details, need to be summarized.
Second, the cited literature supporting proximate composition and fish growth (3.2 section) is quite dated (20-30 years old), I suggest to the authors to add more recent citations.
Finally, just a few specific comments:
Pg.2, line 36: RBT= rainbow trout? The abbreviation is not presented before.
Pg.4, line 22-29: Are the experimental diets designed to be ISO-proteic, ISO-lipidic and ISO-energetic?
Pg.4, line 33 and pg.10 line 35: It is necessary to specify if the time is in a 12-hour or 24-hour clock format.
Pg.5, table 1:
· please change “astazanthin” with “astaxanthin”;
· crude protein percentage is missing in the table;
· Why are other parameters like carbohydrates, fiber etc. not included in the table1?
Pg.9, table 3:
Were the resolution and accuracy of each single probe, placed into the multiparameter sonde, considered from the authors during the data analysis? Could be the instrumental error the most of variability observed in some variables? Check the manufacturer’s specification of Ammonia and Turbidity probes.
Pg.13, table 5: Consider moving the data from table 5 to table 1;
Pg 14, line 4: Do you mean that the initial body weight is not balanced between the experimental tanks? Did you perform a statistical analysis on fish starting size?
Pg 15, table 6:
· initial and final fish weight should be added to the table.
· Survival is presented in M&M as RPS (relative percent survival).
· HCT= haematocrit level? The abbreviation is not presented before in the text as well as the measuring method in M&M; moreover these data are significantly different, but explanation is not given in result section.
Author Response

(The authors gave the same response as above.)

Reviewer 3 Report
Review
Journal Animals (ISSN 2076-2615)
Manuscript ID: animals-2590750
Type: Article
Title: Effect of Dietary Soy Protein Source on Effluent Water Quality and Growth Performance of Rainbow Trout Reared in a Serial Reuse Water System
Authors: Thomas L Welker * , Ken Overturf
Section: Aquatic Animals
General comments and impressions
In general, it is a good paper, well-written, well-planned, and with a great purpose that is to find out alternatives to fishmeal as a main protein source for fish feeding. I think climate-change and overpopulation is pushing us to go further in these initiatives like this one to ensure a sustainable future.
Still I feel that there are some things that should be change in order to make it easier/quicker to understand and more enjoyable paper. I split it within the different sections:
Abstract
- I would recommend to put which genes are you assessing.
- I feel confusing all the abbreviations about the food treatments and very little intuitive, I would rather simplify them as abbreviations like: soy protein concentrated: SP, fermented soy protein (fSP), Fishmeal control (C or C-), high soybean meal (HiS). I meant to clearly identify which group is which.
- RBT I did not find what it is within the text. Is it Rainbow trout? Then, please at least name it once after it. Ex: rainbow trout (RBT). I felt it confusing, especially when you work on fish but not with this specie.
Introduction
- At the sentence ‘Fermentation of soy products can also result in improved nutrient profiles by removing non-nutritive or anti-nutritive components.’ Please, name some examples of non-nutritive and anti-nutritive.
- At the sentence ‘Elevated levels of dietary SBM can cause alterations in the distal intestinal epithelium, leading to pathological changes characterized by diarrhea’. You could also check the inflammatory effect from SBM treatment.
- ‘RNA-sequencing and through the evaluation of expression of known stressor gene’. Please specify which ones. I say this because I am more familiarized by immune responses related with stress, such as inflammatory responses. And I am surprised that you did not check any of those genes such as: IL1B, IL6, TNFa or INFg. So please, explain which genes and why here. You do not have to explain them all, but at least the most important ones, or the ones you have found interesting results (introducing them here).
- The last paragraph, ‘The Trout Grains Research group (USDA, Agriculture Research Service) in Hagerman, (…) control) on water quality, stress response, and growth.’ I would put it at material and methods (M&M), because you are explaining in what your experiment consists.
I would recommend in this part to simplify it telling that ‘the aim of your experiment is by using your circular circuit of water examine the dietary inclusion of three soy protein sources (soybean meal, soy protein concentrate, and a fermented soy protein concentrate) and FM (control) on water quality, stress response, and growth.’ Or something similar.
M&M
2.1. Experimental tank system and fish stocking
- From the sentence ‘However, due to a water delivery pipe failure, high trout mortality occurred in tanks receiving 5th use water, and this part of the project was terminated before the study conclusion’ . If you did not have results from it I would not tell it (remove it).
- ‘6 gallons per minute’. I feel it unnecessary the official measures are L/min
- At ‘To assess the impact of different protein sources on water quality, the water entering each bank of 12 tanks (one for each water source) and exiting each individual tank was continuously monitored. This was achieved using 6920 V2 water quality (…) treatment-water source combination.’ I would simplify it.
2.2. Experimental feeds and feeding
- Here is the same as above, I would use a simpler abbreviation from feed-treatments.
- The sentence ‘apparent feed conversion (FCR)’: means feed conversion rate?
- Time/hours, I do not feel necessary say the time, you can just say that the fish were feed once or twice at day.
- At this sentence ‘There is a possibility some feed was not consumed, which could affect FCR calculations. However, we did not observe a significant amount of uneaten feed during the study’. If by checking/analysing your results they make sense I would not mention it.
- Table 1.
Sodium Chloride is having the double amount in FM-C than the rest of feed treatments
Magnesium oxide is having 10 times more in FM-C than the rest of the feed treatments
I am wondering if that may affect to the results among the treatments more than the fact of the origin of the protein and the amount. If not just justify how it must not affect.
2.3. Fish sampling and physiological analyses
- I feel it confusing. So, how many fish did you sample in total at each time point? I would suggest to put a table instead of this paragraph like: nº of tanks, nº fish/tank, times sampled (you mentioned every 3 weeks until 12 w), which type of samples each time you sampled (like: whole body, liver, gills, sera).
- I have a suggestion for next time, when you want to see immune responses in mucosal surfaces, beside gills you can also check head-kidney, spleen or even the intestine can be useful.
RESULTS and Discussion
3.1. Water quality
- The Snake River Farm is situated in the south-central region of Idaho, along the middle portion of the Snake River. Water is supplied to the farm by the Eastern Snake Plain Aquifer and operates as a single-pass, flow-through raceway system. The typical flow rate is approximately 0.17 m3/s, and water passes through five raceways before being discharged into the quiescent zone [48]. The initial water quality in these raceway systems is defined by the source spring water (1st use) with subsequent passes through raceways reducing water quality. I would change this paragraph and move it to M&M. Instead I would start the paragraph with ‘In this study, temperature, total dissolved solids (TDS), ammonia (NH3), pH, turbidity, and dissolved oxygen (DO) were measured (Table 3). All parameters increased, except for DO which decreased, after passing through tanks (inflow vs. outflow), and water quality was reduced in 3rd use compared to 1st use water which correlated with worse growth performance within an individual diet (e.g., comparing the SBM diet between 1st and 3rd use water). Diet also had a significant impact on growth. The effects of water source and diet on growth will be explored in greater detail below.
- For the temperatures I would write °C instead of C.
- At this sentence ‘pH values exhibited little variation’ you started with a lowercase letter, I would recommend to rephrase the sentence.
- At the sentence ‘at 0900 h and 0700 h’ I would rewrite it as ‘9 am and 7 am’.
- At the sentence ‘various other dissolved substances present in water’, please name examples not just add the ref.
- At the sentence ‘3rd use water may negatively impact trout performance over a longer period of time’ I would specify how, I cannot be reading all the ref to understand your paper. You must put examples.
- Ranging from 18 to 70 Nephelometric Turbidity Units or NTU) [56]. I think you must have explained this at M&M.
3.2. Proximate composition and growth performance
- Table 4 and table 5. I would suggest to put the values in a graph instead, to make it more visual. Why did you used 6f/condition at table 4 and just 3 at table 5?
- At table 6. At ‘Water Source (WS)’ I did not feel it clear, it is an average of all the feed-treatments at 1st use or 3rd use?And if you want this table could be also in graphs.
- This paragraph ‘Fish meal is a highly digestible, complete protein source that continues to be used as the primary protein source in rainbow trout feeds [3]. However, it is a finite resource and continues to increase in price [64], which will continue to increase the cost of trout feed in the future. Therefore, trout producers and feed manufacturers continue to search for low-cost alternative sustainable protein sources for FM. Soybean meal (SBM) has been the most widely used alternative protein source successfully incorporated into diets of rain- bow trout and other fish species [3, 4]. ‘ Something more simplified should be also in the introduction in order to explain why you are exploring new sources of protein to feed the fish.
- At the sentence ‘Adelizi et al. [17] found that rainbow trout (approximately 35 g) fed ..’ . You meant 35g size?
- At the sentence ‘anti-nutritional factors mentioned’ I would add some of the examples you wrote at the Introduction.
- The sentence ‘Fermentation of soy products can also result in improved nutrient profiles by increasing protein levels and also removing non-nutritive or anti-nutritive components’ I feel it quite similar to the just right before, please mix them both.
- The part ‘However, the level of ammonia and urea excretion increased as SPC increased in the diet in their study. We also observed non- significant increases in ammonia in water of trout fed the SPC or SPC-f diets. There was substantial variation among the ammonia values, and our ability to detect significant differences was likely impaired. Although the SPC-f diet also produced significantly higher TDS, it did not appear to affect growth or survival during the study. In past research, we determined negative effects of SPC on fecal stability with reduced water quality – although many ANFs are removed during process of SBM to SPC, the structural fiber content is little affected [26], which may have contributed to poorer water quality in trout fed these diets.’ I feel it repetitive since you already discussed the increment of ammonia at page 11. If you want to mention its relation with what you are now discussing you can do it using a sentence, so please, simplify it.
- Graphs and tables must be self explanatory, so please add at the bottom of each one the abbreviations you are using at the table.
3.3. Cortisol and stress gene responses
- ‘There was significant variation in plasma cortisol values, and this variation likely prevented identification of any significant differences among the experimental treatment combinations.’ The variation of cortisol in serum are very quick (3 min at the net, with a peak at 15 min) and cortisol levels can be rise by handling the fish, so maybe if for some you took more time in fishing them from the tank and put them at the anaesthesia (MS-222), that could be a reason why you had that much variability within the groups.
- Recommendation for next time, the sentence ‘water displayed sub-cellular alterations attributable to exposure to reduced water quality. This indicates that the trout in 3rd use water were likely undergoing chronic stress.’ I think that gene expression is good to assess the transcription level, but if you say that ‘water displayed sub-cellular alterations’ it would be interesting to have some microscopy data from the tissues that may be affected.
- Table 7. Please represent it as a heatmap (to make it more visual)
- Reading this paragraph I am missing some info about the choice of gills and liver as target tissue for stress conditions.
Author Response

(The authors gave the same response as above.)
